# Electromagnetic-Interference-Shielding Effectiveness of Lyocell-Based Carbon Fabrics Carbonized at Various Temperatures

**DOI:** 10.3390/molecules27175392

**Published:** 2022-08-24

**Authors:** Jihyun Park, Lee Ku Kwac, Hong Gun Kim, Kil-Young Park, Ki Woo Koo, Dong-Hwa Ryu, Hye Kyoung Shin

**Affiliations:** 1Institute of Carbon Technology, Jeonju University, Jeonju 55069, Korea; 2Olive Carbon & Solution Co., Ltd., Jeonju 54853, Korea; 3HYOSUNG Advanced Materials Corporation, Jeonju 54849, Korea

**Keywords:** lyocell, carbon fabric, electrical conductivity, electromagnetic interference shielding materials

## Abstract

Lyocell is a biodegradable filament yarn obtained by directly dissolving cellulose in a mixture of *N*-methylmorpholine-*N*-oxide and a non-toxic solvent. Therefore, herein, lyocell fabrics were employed as eco-friendly carbon-precursor substitutes for use as electromagnetic interference (EMI) shielding materials. First, a lyocell fabric treated with polyacrylamide via electron beam irradiation reported in a previous study to increase carbon yields and tensile strengths was carbonized by heating to 900, 1100, and 1300 °C. The carbonization transformed the fabric into a graphitic crystalline structure, and its electrical conductivity and EMI shielding effectiveness (SE) were enhanced despite the absence of metals. For a single sheet, the electrical conductivities of the lyocell-based carbon fabric samples at the different carbonization temperatures were 3.57, 5.96, and 8.91 S m^−1^, leading to an EMI SE of approximately 18, 35, and 82 dB at 1.5–3.0 GHz, respectively. For three sheets of fabric carbonized at 1300 °C, the electrical conductivity was 10.80 S m^−1^, resulting in an excellent EMI SE of approximately 105 dB. Generally, EM radiation is reduced by 99.9999% in instances when the EMI SE was over 60 dB. The EMI SE of the three lyocell-based carbon fabric sheets obtained at 1100 °C and that of all the sheets of the sample obtained at 1300 °C exceeded approximately 60 dB.

## 1. Introduction

Technological advancements in electronic devices have led to critical electromagnetic complications that adversely impact machine operating life, the environment, and human health [1,2,3,4,5,6]. The extensive study of electromagnetic interference (EMI) materials is necessary to establish preventative measures regarding electromagnetic radiation pollution. Metals are representative additives for EMI shielding owing to their high electrical conductivity. However, the drawbacks of metals, such as their high density, corrosion, and rigidity, impose limitations on their use in flexible electronic devices [7,8,9,10,11,12,13,14]. In recent times, carbon-based materials have been widely introduced as suitable EMI-shielding materials owing to their light-weight, non-corrosiveness, and chemical stability, as well as their high electrical and thermal conductivities. Graphenes, carbon nanotubes (CNTs), graphites, and carbon fibers have been extensively researched in EMI shielding applications by their use as conductive additives in foams, composites, sponges, papers, and aerogel [15,16,17,18,19,20,21,22]. Among them, carbon fibers with high aspect ratios have been widely used as EMI shields; specifically, such materials have mainly been studied for integration in carbon-fiber-reinforced polymers or metal-coated carbon-fiber-reinforced composites obtained after cutting carbon fibers [23,24,25]. However, carbon fabric obtained by weaving carbon fibers can be independently used as an EMI-shielding material, i.e., without other conductive fillers. In addition, carbon fabric has considerable advantages, such as flexibility, which translates to excellent formability. Generally, carbon fabrics can be formed from three precursors, namely polyacrylonitrile (PAN), pitch, and cellulose. PAN is widely used as a precursor for most of the carbon fibers for fabricating high-performance carbon fibers, and pitch-based carbon fibers are extensively used in the fields of application with high Young’s modulus and thermal conductivity. However, PAN and pitch, typical precursors of carbon fabrics, originate from fossil fuels, which impose various limitations owing to the cost of crude oil, toxic gas generation during the carbon-fiber manufacturing process, and secondary pollution generated after use [26,27]. Meanwhile, lyocell is a biodegradable filament yarn obtained through a dry-jet wet spinning method by directly dissolving cellulose in an *N*-methylmorpholine-*N*-oxide (NMMO), non-toxic solvent, or NMMO-water mixture without aging, xanthation, and mercerization treatment. Additionally, lyocell can be continuously and stably supplied owing to the abundance of cellulose on earth and the eco-friendly fabrication process to produce yarn from raw materials [28,29]. Further, when carbon fabrics are prepared using lyocell, secondary pollution can be prevented. In summary, lyocell is considered a supreme material for EMI shielding because it is eco-friendly and flexible. However, lyocell-based carbon fibers have disadvantages associated with longer stabilization times, lower mechanical properties, and lower carbon yields compared to PAN- and pitch-based carbon fibers. Generally, to improve carbon yields and mechanical properties, lyocell-based carbon fabric is prepared via thermal stabilization only [30] or pretreated using flame retardants or a silicon-based polymer before thermal stabilization [31,32]. However, research on the EMI shielding effectiveness of the obtained lyocell-based carbon fabric has not been conducted. In this study, we specifically investigated the EMl-shielding effectiveness (SE) of carbon fabrics obtained from lyocell grafted using polyacrylamide (PAM) via electron-beam irradiation (EBI). Kim et al. [33] prepared carbon fiber obtained from lyocell grafted using a 100 kGy dose of EBI dose according to the PAM concentration. The results revealed a 55% carbon yield of the thermally stabilized lyocell fiber grafted with 0.5 wt.% PAM, along with high tensile strength of 1.39 GPa. The lyocell-based carbon fiber preparation method reported in this study holds several merits: EBI dose treatment of over 1000 kGy was further decreased to 100 kGy upon treating PAM, and the thermal stabilization time was decreased to 1 h compared to that reported in the previous study [34,35,36,37,38], while the tensile strength of over 1 GPa was obtained. With reference to this data, we prepared carbon fabrics obtained through the carbonization of lyocell fabric grafted with 0.5 wt.% PAM using a 100 kGy dose of EBI, which showed good mechanical properties and carbon yield. We subsequently investigated the effect of their lyocell-based carbon fabrics carbonized at various temperatures on EMI SE. 

## 2. Results and Discussion

### 2.1. X-ray Diffraction and Raman Spectra Studies of Lyocell-Based Carbon Fabrics

Figure 1a exhibits the XRD profiles of lyocell-based carbon fabrics obtained through various carbonization temperatures. All lyocell-based carbon fabrics displayed broad peaks at 24 and 43°, associated with the (002) and (100) planes, respectively. Notably, (002) lattice diffraction peak patterns were clear. These two peaks at 24 and 43° indicate the development of the graphitic crystalline structures [15,19,33,34,35,36,37,38]. In Figure 1a, with an increase in carbonization temperature to 900 °C, 1100 °C, and 1300 °C, the full width at half-maximum (FWHM) values at 2*θ* values of 24° decreased to 8.09, 7.28, and 6.59°, while the intensities of these respective peaks increased. These results demonstrated that lyocell-based carbon fabrics transformed into higher-degree graphitic crystalline structures upon increasing the carbonization temperature. To comprehend the conversion degree of the graphitic structure of the lyocell-based carbon fabrics obtained at the various carbonization temperatures, the Raman spectra of the sample were measured, which are shown in Figure 1b. As shown, the two peaks at 1351 and 1610 cm^−1^ were observed in all samples. The characteristic peak at 1351 cm^−1^ is related to the D-band, which describes the vibrations in the amorphous region, while the peak at 1610 cm^−1^ corresponds to vibrations corresponding to the ordered graphitic region for sp^2^ carbons. As the carbonization temperature increased, the intensity of the G band increased more than that of the D band, and thus, the peak ratios of the G/D bands (I_G_/I_D_) gradually increased. These results indicated that the graphitic crystalline structures expanded with the increase in carbonization temperature.

### 2.2. Electrical Conductivity and Electromagnetic Interference Shielding Effectiveness

Figure 2a displays the electrical conductivity of lyocell-based carbon fabrics according to various carbonization temperatures and the number of stacked carbon fabric sheets. With an increase in the carbonization temperature and the number of stacked carbon fabric sheets, the electrical conductivity increased. For example, for a sing sheet of a lyocell-based carbon fabric obtained by carbonization at 900, 1100, and 1300 °C, the electrical conductivity obtained were 3.57 ± 0.15, 5.96 ± 0.31, and 8.91 ± 0.24 S m^−1^, respectively. This is due to the growth of the crystallographic order of carbon fabrics in line with the carbonization temperature that facilitated electron transport [39,40,41]. Finally, when three sheets of lyocell-based carbon fabrics obtained at 1300 °C were stacked, the highest electrical conductivity, 10.80 ± 0.30 S m^−1^, was obtained. These high electrical conductivities were om accordance with the EMI SE effects. Additional evidence of the high electrical conductivity of the lyocell-based carbon fabrics is shown in Figure 2b, which illustrates a lyocell-based carbon fabric lighting up an electric bulb in a closed loop. 

Generally, the total EMI SE values can be obtained as a comprehensible sum by determining the reflection, absorption, and multiple reflection damages of electromagnetic radiation on a shielding material. When the surface of the shielding material is exposed to EM radiation, the radiation is decreased owing to its reflection because of the impedance discrepancy between two dissimilar mediums, namely the shielding materials and space, while the unreflected part is convulsively reduced through absorption by the shielding material. Additionally, when the un-reflected EM radiation reaches the end of the shielding material, partial radiation is transmitted into the material, and the remainder is reflected, resulting in a continuous decrease in EM radiation by multiple reflections within EM materials. A schematic diagram for the EMI shielding mechanism is represented in Figure 3. Therefore, the total SE can be calculated using the Schelkunoff theorem per the following formula [42,43]:R = |S_11_|^2^, T = |S_21_|^2^, A + R + T = 1
SE_A_ = −10 log (T/(1 − R)), SE_R_ = −10 log (1 − R)
SE_T_ = SE_A_ + SE_R_ + SE_M_
where R is reflectivity, T is transmittance, A is absorptivity, SET is the total shielding effectiveness, SE_A_ is absorption shielding effectiveness, SE_R_ is reflection shielding effectiveness, and SE_M_ is multiple reflection. Generally, SE_M_ is disregarded in the event that the total SE is over10 dB.
SE_T_ ≈ SE_A_ + SE_R_ = −10 log T= −S_21_

Figure 4 exhibits the results of EMI SE of lyocell-based carbon fabrics according to various carbonization temperatures and numbers of stacked carbon fabric sheets. Typically, EM shielding materials have high electrical conductivity. As shown in Figure 4, EMI SE increased as carbonization temperature increased due to the increased electrical conductivity. In the case where the total EMI SE was over 60 dB, EM radiation was reduced by 99.9999% [44]. In the case of three sheets of lyocell-based carbon fabric carbonized at 1100 °C, the EMI SE was over 60 dB in the range of 1.5–3.0 GHz. In the case of all samples carbonized at 1300 °C, the EMI SE exceeded 80 dB in all measured ranges. Notably, in the case of three sheets of lyocell-based carbon fabric carbonized at 1300 °C, the EMI SE was the highest at approximately 105 dB. Compared to other previous studies, Yim et al. [25] reported on the EMI SE of electroless FeCoNi-plated carbon fibers. Although three different kinds of metals were plated on the carbon fiber, per the results, an EMI SE of 69.4 dB at 1.5 GHz was observed for 60-FeCoNi-plated carbon fibers treated for 60 min in the plating solution. Additionally, Li et al. [45] fabricated carbon fabric-NiCo composites, and it was observed that a carbon fabric-NiCo_2_O_4_ composite, having a thickness of 0.34 mm, resulted in an EMI SE of 53 dB. Further, Thi et al. [46] researched flexible MOF on Co_X_Fe_1−__X_OOH@carbon films for sufficient EMI shielding, and it was observed that the EMI SE of CoFe/CoCu-carbon films (MOF growing for 45 min) was 73.46 dB. Lastly, Park et al. [19] investigated the EMI SE of carbon papers obtained from blending tall goldenrod celluloses and carbon nanotubes (CNTs), and it was reported that cellulose carbon paper containing 15 wt% CNTs with a thickness of 4.5 mm, carbonized at 1300 °C, showed an EMI SE of 62 dB at 1.6 GHz. Therefore, in the case of the lyocell-based carbon fabrics carbonized at 1300 °C, an excellent EMI SE exceeding 80 dB was observed without plating or adding any metals or carbonaceous fillers. 

### 2.3. Morphology and Mechanical Properties of Lyocell-Based Carbon Fabric

Figure 5 shows surface and cross-sectional images of the lyocell-based carbon fabric carbonized at 1300 °C. As shown in Figure 5a, the twill shape of the fabric is uniformly maintained without cracks or porosities even after carbonization temperature at 1300 °C. In Figure 5b, carbon fibers plucked from the carbon fabric appear to have smooth surfaces and clearly circular cross-sections. Figure 6 displays the estimated tensile strengths of fibers plucked from the lyocell-based carbon fabric carbonized at various temperatures. The tensile strengths of the carbon fibers for all carbonization temperatures were over 1.3 GPa, with a slight increase observed in carbon fibers with a higher carbonization temperature owing to the expansion of the ordered graphitic region. 

## 3. Materials and Methods

### 3.1. Materials

Lyocell fabrics that were composed of 900 filaments of 1650 d (use of conifer wood pulp obtained from Southern Pine), were acquired from Hyosung Co. (Ulsan, Korea), and PAM was supplied by Sigma-Aldrich (St. Louis, MO, USA).

### 3.2. Preparation of Lyocell-Based Barbon Fabrics

In order to obtain lyocell-based carbon fabrics, lyocell fabrics (measuring 10 × 5 cm^2^) were immersed in 0.5 wt. % PAM solution and irradiated with a 100 kGy. EBI management was performed using a conveyor-style scanned beam at an accelerating voltage of 1.14 MeV, a beam current of 7.6 mA, and a dosage rate of 6.67 kGy/s in the air at room temperature. Lyocell fabrics treated with EBI in 0.5 wt. % PAM solution were lyophilized and then thermally treated at 300 °C for 1 h. Subsequently, carbonization was conducted at 900, 1100, and 1300 °C without carbonization, keeping time in a tubular furnace with a pure nitrogen gas (99.999%) atmosphere. Figure 7 shows a schematic illustration for the carbonized lyocell fabrics according to carbonization temperature.

### 3.3. Characterization

In order to estimate the crystallinity of lyocell-based carbon fabrics obtained through various carbonization temperatures, XRD profiles were recorded using D/MAX-2500 (RIGAKU, Tokyo, Japan) at 40 kV and 30 mA with CuKα radiation. Raman spectra were accomplished on an ARAMIS instrument (Horiba Jobin Yvon, Tokyo, Japan) with a 514 nm laser in the range from 500 to 3500 cm^−1^. The electrical conductivity was obtained by measuring the surface and volume resistivities of samples using MCP-1700 LOTESTA-GX (NITTOSEIKO ANALYTECH, Kanagawa, Japan). The EMI SE was analyzed using E5071C ENA Vector Network Analyzer (Keysight, Santan Rosa, CA, USA) in the frequency range from 0.5 to 1.6 GHz. Figure 8 illustrates the EMI shielding analyzer. The tensile strengths of lyocell-based carbon fabrics for various carbonization temperatures were measured using an Automatic Single-Fiber Tester Favigraph (Textechno, Mönchengladbach, Germany) equipped with a load cell of 100 cN after plucking a single fiber from the carbon fabric. 

## 4. Conclusions

Lyocell fabrics treated with EBI and PAM were carbonized at 900, 1100, and 1300 °C. The obtained lyocell-based carbon fabrics were applied as EMI shielding materials. First, XRD and Raman analyses showed that an increase in carbonization temperature entailed an increase in the graphitic crystalline structure. This enhanced the electrical conductivity and EMI SE of the lyocell-based carbon fabrics. The electrical conductivity of single-layer lyocell-based carbon fabrics was 3.57, 5.96, and 8.91 S m^−1^. As the thickness of the fabrics increased, the electrical conductivity increased further. When three sheets of lyocell-based carbon fabrics carbonized at 1300 °C were stacked, the electrical conductivity reached 10.80 S m^−1^. These high electrical conductivities promoted the EMI SE. In the case of three sheets of lyocell-based carbon fabric carbonized at 1100 °C, the EMI SE was over 60 dB, corresponding to the reduction of 99.9999% of EM radiation in the range of 1.5–3.0 GHz. Moreover, samples obtained at 1300 °C, the highest carbonation temperature in this study, showed EMI SEs of 80 dB to 105 dB in all measured ranges. Such lyocell-based carbon fabrics with high electrical conductivity and EMI SE can be used in EMI shielding applications owing to their flexibility and light weight. They can be translated into various shapes due to their excellent formability. However, it is necessary to investigate the EMI SE of lyocell-based carbon fabrics carbonized at temperatures over 1300 °C in the future.

## Figures and Tables

**Figure 1 molecules-27-05392-f001:**
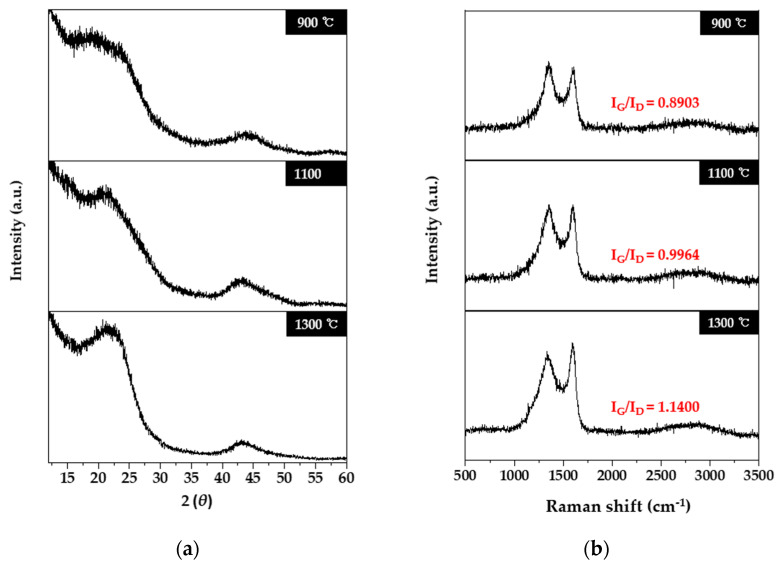
X-ray diffraction (XRD) patterns (**a**) and Raman spectra (**b**) of lyocell-based carbon fabrics obtained at various carbonization temperatures.

**Figure 2 molecules-27-05392-f002:**
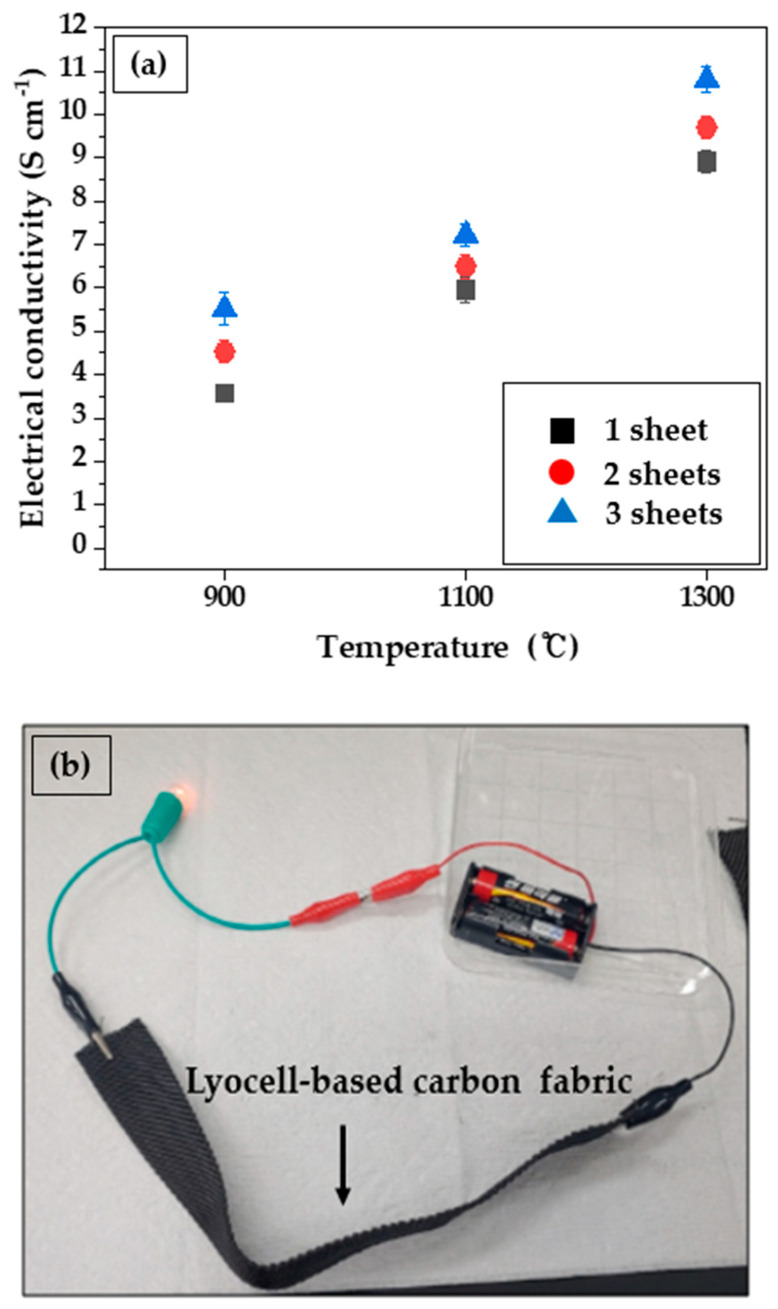
Electrical conductivity of lyocell-based carbon fabrics according to various carbonization temperature (**a**) and number of stacked carbon fabric sheets (**b**).

**Figure 3 molecules-27-05392-f003:**
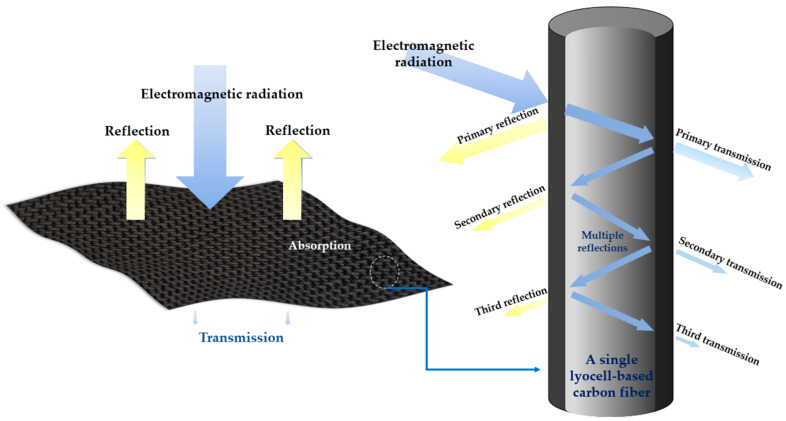
A schematic diagram of electromagnetic interference (EMI) shielding mechanism.

**Figure 4 molecules-27-05392-f004:**
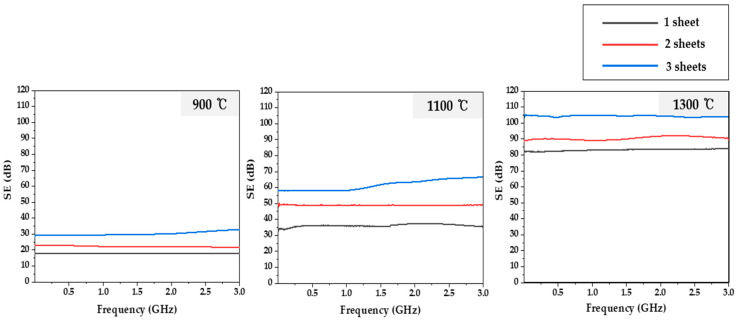
Electromagnetic interference shielding effectiveness (EMI SE) of lyocell-based carbon fabrics according to various carbonization temperatures and number of stacked carbon fabric sheets.

**Figure 5 molecules-27-05392-f005:**
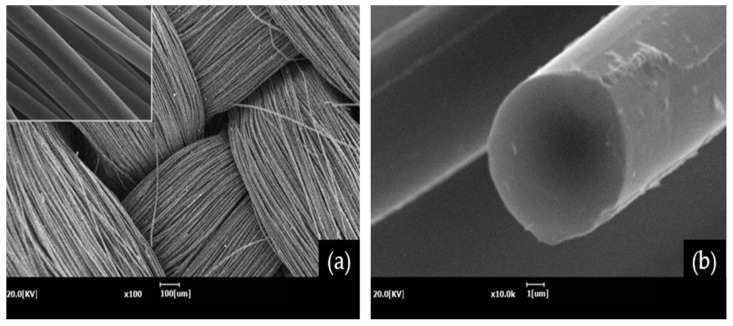
SEM images of surface of (**a**) lyocell-based carbon fabric and (**b**) its single fiber.

**Figure 6 molecules-27-05392-f006:**
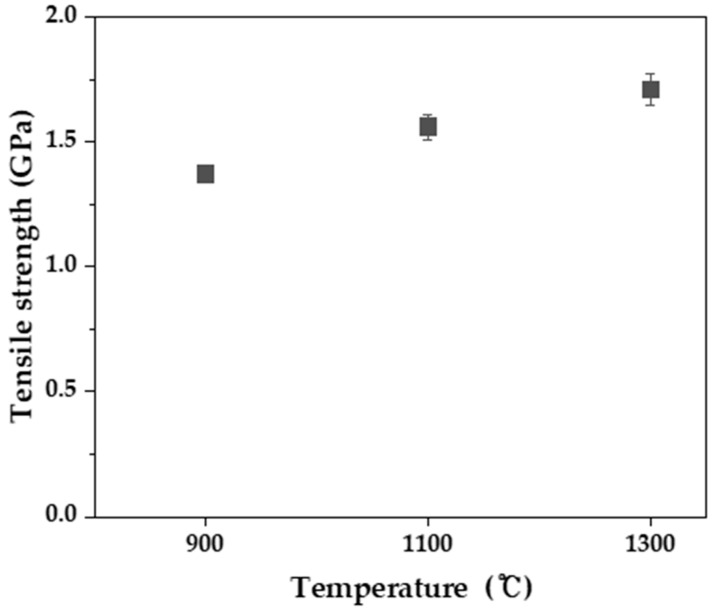
Tensile strengths of a single fiber plucked from a lyocell-based carbon fabric for various carbonization temperatures.

**Figure 7 molecules-27-05392-f007:**
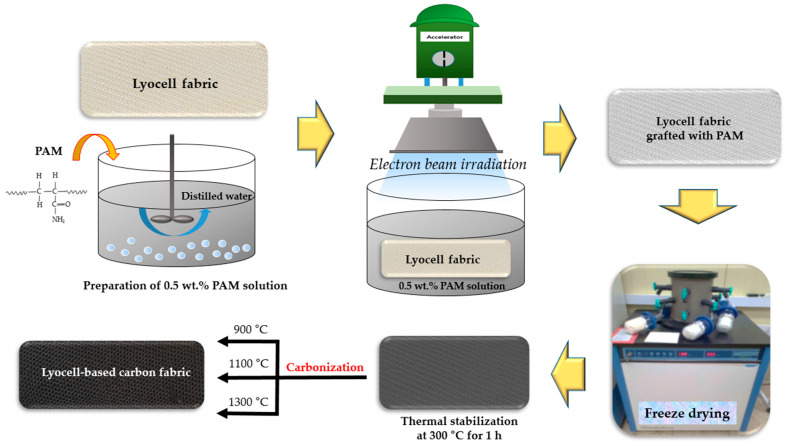
Schematic for the preparation of lyocell-based carbon fabric.

**Figure 8 molecules-27-05392-f008:**
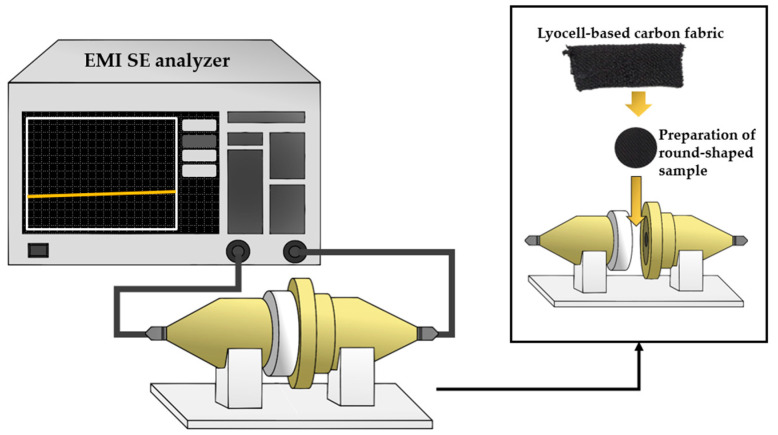
Schematic diagram of electromagnetic interference shielding effectiveness apparatus.

## Data Availability

Not applicable.

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
