# Peer review of "Electromagnetic-Interference-Shielding Effectiveness of Lyocell-Based Carbon Fabrics Carbonized at Various Temperatures"

_molecules, 2022, doi:10.3390/molecules27175392_

Round 1

Reviewer 1 Report

In this article, the authors studied Electromagnetic-Interference-Shielding Effectiveness of 2 Lyocell-based Carbon Fabrics Carbonized at Various 3 Temperatures. The article could be acceptable for publication but it needs some revisions to help the authors and ensure the quality of the published papers in this journal.

1.      The whole manuscript must be carefully revised for sentence construction and language errors (certificate for the proofreading should be provided).

2.      Please re-write the abstract with brief overview of the preparation, emphasizing results and purpose of study.  What is EBI and PAM? Please define all abbreviations.

3.      There is no much discussion, the results in this study has been described extensively. The purpose of this section is to interpret, describe and compare the significance of your finding in light of what was already known about the research problem being investigated. The cited reference in this section is only 4 out of 37. Therefore, the obtained results should have compared with previous/published results

4.      In the discussion section, the authors should talk about potential errors from the machine resolution for the experiments and include errors bars in their results.

5.      The research problem should be stated concisely and identify research gaps.

6.      What is the time of carbonization?

7.      What is the tensile test condition and sample size and standard used?

8.      A schematic diagram for the EMI shielding mechanism needs more details in the introduction with include related Refs.

9.      The authors stated that To estimate the crystallinity of lyocell-based carbon fabrics obtained through various carbonization temperatures, XRD profiles were recorded But there is not any crystallinity values? Please explain.

10.   Please provide stress-strain curve for the tensile strength test.

Author Response

Dear reviewer

We are thankful for your constructive comments on this manuscript, and have made the following revisions in response to them.

  1. The whole manuscript must be carefully revised for sentence construction and language errors (certificate for the proofreading should be provided).

I have revised the entire manuscript to address issues pertaining to sentence construction, language, and grammar. Moreover, the language of the paper was reviewed by a prominent journal proofreading organization, named Editage. Please find attached the certificate for the same.

  1. Please re-write the abstract with brief overview of the preparation, emphasizing results and purpose of study. What is EBI and PAM? Please define all abbreviations.

 We have revised the abstract per your comments

After correction: Lyocell is a biodegradable filament yarn obtained by directly dissolving cellulose in a mixture of N-methylmorpholine-N-oxide and a non-toxic solvent. Therefore, herein, lyocell fabrics were employed as eco-friendly carbon-precursor substitutes for use as electromagnetic interference (EMI) shielding materials. First, a lyocell fabric treated with polyacrylamide via electron beam irradiation reported in a previous study, to increase carbon yields and tensile strengths was carbonized by heating to 900, 1100, and 1300 °C. The carbonization transformed the fabric into graphitic crystalline structure, and its electrical conductivity and EMI shielding effectiveness (SE) were enhanced despite the absence of metals. For a single sheet, the electrical conductivities of the lyocell-based carbon fabric samples at the different carbonization temperatures were 3.57, 5.96, and 8.91 S m-1, leading to an EMI SE of approximately 18, 35, and 82 dB at 1.5–3.0 GHz, respectively. For three sheets of the fabric carbonized at 1300 °C, the electrical conductivity was 10.80 S m-1, resulting in an excellent EMI SE of approximately 105 dB. Generally, EM radiation is reduced by 99.9999% in instances when the EMI SE was over 60 dB. The EMI SE of the three lyocell-based carbon fabric sheets obtained at 1100 °C and that of all the sheets of the sample obtained at 1300 °C exceeded approximately 60 dB.

  1. There is no much discussion, the results in this study has been described extensively. The purpose of this section is to interpret, describe and compare the significance of your finding in light of what was already known about the research problem being investigated. The cited reference in this section is only 4 out of 37. Therefore, the obtained results should have compared with previous/published results

Per your comments, we have added the following sentences:

“ Compared to other previous studies, Yim et al. [25] reported on the EMI SE of electroless FeCoNi-plated carbon fibers. Although three different kinds of metals were plated on the carbon fiber, per the results, an EMI SE of 69.4 dB at 1.5 GHz was observed for 60- FeCoNi-plated carbon fibers treated for 60min in plating solution. Additionally, Li et al. [45] fabricated carbon fabric-NiCo composites, and it was observed that a carbon fabric-NiCo2O4 composite having a thickness of 0.34 mm resulted in an EMI SE of 53 dB. Further, Thi et al. [46] researched on flexible MOF on CoXFe1-XOOH@carbon films for sufficient EMI shielding, and it was observed that the EMI SE of CoFe/CoCu-carbon films (MOF growing for 45min) was 73.46 dB. Lastly, Park et al. [19] investigated the EMI SE of carbon papers obtained from blending tall goldenrod celluloses and carbon nanotubes (CNTs), and it was reported that cellulose carbon paper containing 15 wt% CNTs with a thickness of 4.5 mm, carbonized at 1300 °C, showed an EMI SE of 62 dB at 1.6 GHz. Therefore, in the case of the lyocell-based carbon fabrics carbonized at 1300 °C, an excellent EMI SE exceeding 80 dB was observed without plating or adding any metals or carbonaceous fillers.”

  1. In the discussion section, the authors should talk about potential errors from the machine resolution for the experiments and include errors bars in their results.

    The error bars have been added in the revised paper.

  1. The research problem should be stated concisely and identify research gaps.

Per your comments, in line238-239, the following statement was added:

However, it is necessary to investigate the EMI SE of lyocell-based carbon fabrics carbonized at temperatures over 1300 °C in the future.

  1. What is the time of carbonization?

  The Carbonization keeping time was 0 min. Thus, I added the following phrase “without carbonization keeping time” to the text for clarity.

  1. What is the tensile test condition and sample size and standard used?

 The tensile test condition was explained under the section “3.3. Characterization”.

  1. A schematic diagram for the EMI shielding mechanism needs more details in the introduction with include related Refs.

 We had already explained this in section “2.2. Electrical Conductivity and Electromagnetic Interference Shielding effectiveness”. However, the following statements were added:

Therefore, the total SE can be calculated using the Schelkunoff theorem per the following formula [43, 44 ]:

R=|S11|2,  T=|S21|2,  A+R+T=1

SEA = -10 log (T/(1-R)), SER= -10 log (1-R)

SET = SEA + SER+ SEM

where R is reflectivity, T is tansmittance, A is absorptivity, SET is the total shielding effectiveness, SEA is absorption shielding effectiveness, SER is reflection shielding effectiveness, and SEM is multiple reflection. Generally, SEM is disregarded in the event that the total SE is over10 dB.

SET ≈ SEA + SER = - 10 log T= - S21

  1. The authors stated that To estimate the crystallinity of lyocell-based carbon fabrics obtained through various carbonization temperatures, XRD profiles were recorded But there is not any crystallinity values? Please explain.

      As mentioned in the XRD profiles, the two peaks at 24 and 43° indicate the development of the graphitic crystalline structures. The width of the peak at 24° of the main peak (FWHM) narrowed, which was regarded as the development of the graphitic crystalline structures and not the crystallinity values. The relevant references had been added to the text [15, 19, 33, 38]

  1. Please provide stress-strain curve for the tensile strength test.

        Per your comments, I have revised the figure to show the stress-strain curves.

Thank you again for your valuable comments and insightful suggestions.

Best regards.

Dr. Hye Kyoung Shin

Reviewer 2 Report

The article titled “Electromagnetic-Interference-Shielding Effectiveness of  Lyocell-based Carbon Fabrics Carbonized at Various Temperatures” is devoted to the study of lyocell fabrics as EMI shields. In this aspect, the article is interesting and well written.

However, there are a limited number of concerns that need some improvements, and that are listed below:

1.       Please, in the abstract section, define EBI and PAM. Note that acronyms must be defined the first time they appear in the text.

2.       In the introduction section, the authors make a series of assert respecting the disadvantages of PAN and pitch as typical precursors of carbon fabric since they (as coming from fossil fuel) have a series of limitations including the fluctuation in cost. However, this reviewer has not read in this section any information related to the much more expensive cost of the lyocell-based materials respecting the classical ones. So, being honest, this information must be also included in the introduction to avoid the false idea of the suggested lyocell as a panacea precursor.

3.       In the same contest, this reviewer would suggest including a comparison with other carbon fabrics to check the real technical advantages of Lyocell-based fabrics over conventional carbon fabrics. A bibliographic comparison would be enough.

4.       In the materials section, very poor information about the Lyondell fabrics is provided. Please provide some information rather than the mere supplier. Note that Lyocell is a rayon-like semi-synthetic fiber obtained from a series of wood pulp coming from some sources (oak, bamboo, eucalyptus, et cetera) and are several grades in the market according to the fineness (Dtex) and length (standard, micro, LF, A100, Fill,…). So, the authors should include some of this information, at least the grade used to manufacture the fabrics. Please, note that this is important for traceability reasons, in the realm of scientific communication.

5.       The conclusion section is just a resume of the obtained data but fails in highlighting the advantages of using this kind of fabric. Please, try to provide this critical information.

In the light of the above concerns, this reviewer would like to recommend a MAJOR revision.

Author Response

Dear reviewer

We are thankful for your constructive comments on this manuscript, and have made the following revisions in response to them.

  1. Please, in the abstract section, define EBI and PAM. Note that acronyms must be defined the first time they appear in the text.

→ We have revised “EBI and PAM” and defined the abbreviations as “electron beam irradiation (EBI) and polyacryamide (PAM)” in line 14 per your comments

  1. In the introduction section, the authors make a series of assert respecting the disadvantages of PAN and pitch as typical precursors of carbon fabric since they (as coming from fossil fuel) have a series of limitations including the fluctuation in cost. However, this reviewer has not read in this section any information related to the much more expensive cost of the lyocell-based materials respecting the classical ones. So, being honest, this information must be also included in the introduction to avoid the false idea of the suggested lyocell as a panacea precursor.

→ We have deleted the phrase “such as cost fluctuations” and added the following passage instead;

“PAN is widely used as a precursor for most of the carbon fibers for fabricating high-performance carbon fibers and pitch-based carbon fibers are extensively used in the fields of application with high Young’s modulus and thermal conductivity.” “However, lyocell-based carbon fibers have disadvantages associated with longer stabilization times, lower mechanical properties, and lower carbon yields, compared to PAN- and pitch-based carbon fibers.”

This revision was made to avoid the false idea of the suggested lyocell as a panacea precursor.

  1. In the same contest, this reviewer would suggest including a comparison with other carbon fabrics to check the real technical advantages of Lyocell-based fabrics over conventional carbon fabrics. A bibliographic comparison would be enough.

→ Per your comments, we have added the following sentence:

 “The lyocell-based carbon fiber preparation method reported in this study holds several merits: EBI dose treatment of over 1000 kGy was further decreased to 100 kGy upon treating PAM, and the thermal stabilization time was decreased to 1 h compared to that reported in the previous study [34-38], while the tensile strength of over 1 GPa was obtained.”

  1. In the materials section, very poor information about the Lyondell fabrics is provided. Please provide some information rather than the mere supplier. Note that Lyocell is a rayon-like semi-synthetic fiber obtained from a series of wood pulp coming from some sources (oak, bamboo, eucalyptus, et cetera) and are several grades in the market according to the fineness (Dtex) and length (standard, micro, LF, A100, Fill,…). So, the authors should include some of this information, at least the grade used to manufacture the fabrics. Please, note that this is important for traceability reasons, in the realm of scientific communication.

→ Per your comments, I have provided certain basic information on the lyocell. However, we are unable to provide any further information at this point due to organizational confidentiality reasons. I hope you’d understand.

After correction: Lyocell fabrics which was composed of 900 filaments of 1650 d (use of conifer wood pulp obtained from Southern Pine) were acquired from Hyosung Co. (Ulsan, Korea)

  1. The conclusion section is just a resume of the obtained data but fails in highlighting the advantages of using this kind of fabric. Please, try to provide this critical information.

→ I have added the following sentence in the “Conclusion” section:

Such lyocell-based carbon fabrics with high electrical conductivity and EMI SE can be used in EMI shielding applications owing to their flexibility and light weight. They can be translated into various shapes due to excellent formability.

Thank you again for your valuable comments and insightful suggestions.

Best regards.

Dr. Hye Kyoung Shin

Round 2

Reviewer 2 Report

The authors have performed fine revision work by providing convincing arguments. Consequently, this reviewer can recommend the acceptation of the paper in its actual stage.